# A Centralized Multi-User Anti-Composite Intelligent Interference Algorithm Based on Improved Q-Learning

**Yingtao Niu** [1] , **Boyu Wan** [2],*** **and Changxing Chen** [2]

1    The Sixty-Third Research Institute, National University of Defense Technology, Nanjing 210007, China
2    Fundamentals Department, Air Force Engineering University of PLA, Xi'an 710051, China
*    Correspondence: wby1196874871@163.com

**Abstract:** This paper proposes a central anti-jamming algorithm (CAJA) based on improved Q-learning to further solve the communication challenges faced by multi-user wireless communication networks in terms of external complex malicious interference. This will also reduce the dual factors restricting wireless communication quality, the impact of inter-user interference within the network, and the effect of external malicious interference on the communication system to improve multi-user wireless communication transmission. Firstly, a central base station that coordinates and allocates channels for users within the network is set up using multi-user wireless communication network architecture to constitute a centralized wireless communication network. Secondly, the multi-user system is modeled using the single-user Markov decision process in which the central base station is the main body. Finally, an improved Q-learning algorithm is used to improve overall system transmission income using the central base station, based on the network user number sequential decision action for avoiding external malicious interference. It is designed to avoid the impact of internal network interference on transmission performance during the early stage of communication, achieving overall system transmission income improvement. Simulation results show that in comparison to the existing multi-user independent Q-learning anti-jamming algorithm and the traditional orthogonal frequency-hopping scheme, the proposed algorithm significantly improves overall system transmission performance.

**Keywords:** Q learning; compound intelligent interference; multi-user; centralized wireless communication network





## 1. Introduction

As the advancement of communication equipment continues, wireless communication has developed in a networked manner [1,2]. A wireless sensor network composed of numerous multifunctional sensor nodes is a major example of networking development within the field of sensor application, and it has been applied in many fields [3,4]. However, due to the open nature of wireless communication networks and a challenging and increasingly complex work environment, it can easily be affected by external malicious interference. This significantly impacts the dependability and efficiency of wireless communication in a negative way. More specifically, the communication transmission of the system is blocked, resulting in subpar overall system transmission performance. Internal network factors and external environment aspects must be considered equally. As the number of network users increases, the influence of mutual disturbance on them becomes of increasing significance. Therefore, dealing with external malicious interference and the interference between users inside the network, ensuring the transmission performance of the infinite communication network, and realizing the reliable and effective transmission of the network in an effective manner are urgent issues that must be solved.

Traditional communication anti-jamming technology is mainly based on spread spectrum communication anti-jamming technology, including frequency hopping spread spec-

trum (FHSS) [5] and direct sequence spread spectrum (DSSS) [6]. Conventional communication anti-jamming techniques can help deal with conventional interference in an effective manner. However, as alternative communication interference and communication anti-jamming technology research advances, guaranteeing the dependability and effectiveness of the communication transmission of the system in the face of malicious interference with diversity, dynamics, and other intelligent characteristics is quite difficult [7].

Machine learning is a technical means for transforming the collected external data into its own application knowledge base [8], providing a feasible new approach for multi-user wireless communication networks for coping with external intelligent malicious interference and user internal interference [9,10].

By relying on user actions in the network and feedback from the outside environment, reinforcement learning, which is an indispensable subfield of machine learning, can be used for constructing a model that is not based on interference itself [11]. The method of "searching for optimal solutions through errors" can then be used to adjust the choice of actions in the feedback from the environment as a means of realizing intelligent anti-jamming communication for the network. Reinforcement learning algorithms that are applied to multi-user wireless communication networks can be categorized as distributed anti-jamming [3,12–14] and central anti-jamming [15,16], based on different decision subjects. With distributed anti-jamming, the subject that performs anti-jamming decisions through the algorithm is each user in the network. Yao et al. [3] proposed a collaborative anti-jamming algorithm for multi-user networks, but the internal users are tiny, and the external interference is sweep interference, which means that the interference has strong regularity and can easily be mastered by the communication party. Lowe et al. [12] proposed an actor–critic-based algorithm that is applicable to multi-user communication networks and fully considers the interference between users, effectively improving the transmission performance of the system. However, it lacks consideration of external malicious interference. Zhang et al. [13] adopted the double Q-learning algorithm for effectively dealing with the constant interference of power tracking. This can improve the maximum throughput of a multi-user wireless communication network. Zhang et al. [14] proposed a model-based Dyna-Q learning algorithm for distributed multi-user systems. By transmitting data in the jamming environment from the source to the host, Dyna-Q selects appropriate relay nodes for maintaining multi-user wireless communication network communication, thereby improving convergence speed. Wang et al. [17] used a model-free reinforcement learning algorithm (MRL), combined with the concept of mean field, and proposed a mean field Q-learning algorithm, which can effectively solve large-scale network communication problems; Yao et al. [18] modeled the multi-user anti-interference channel intervention problem of fully connected peer-to-peer networks as a Markov game. They also proposed a multi-intelligence Q-learning anti-interference channel access algorithm based on multi-intelligence, which can enable intra-network users to learn jointly by avoiding inter-user bypassing and external sweep interference. However, the subject of the central anti-jamming decision is the central base station in the network, which is responsible for the coordination and distribution of actions for each user. Aref et al. [15] designed a multi-user cooperative anti-jamming algorithm for the central network, but it has a poor convergence effect. Zhou et al. [16] proposed an anti-jamming scheme that is based on multi-user joint Q-learning. However, network user disturbance during the early stages of communication cannot yet be avoided. For effectively dealing with intelligent malicious interference, reducing the influence of interference between users in the initial communication network and realizing the reliable transmission of a multi-user wireless communication network, a central base station is added in this paper. This enables the coordination and activation of access channels for internal users on the system architecture of a loosely coupled multi-user wireless communication network. At the same time, a central anti-jamming algorithm is proposed that is based on improved Q-learning. This paper makes the following contributions:

A centralized multi-user anti-jamming algorithm based on improved Q-learning that models the system model using a single-user Markov decision process is proposed for a

multi-user wireless communication system with an additional central base station as the decision subject. This will be used to deal with the multi-user wireless network intelligent anti-jamming communication problem. Simulation results find that the proposed algorithm has the ability to effectively improve multi-user wireless communication network system transmission income.

The paper is arranged as follows: Section 2 focuses on the modeling of multi-user wireless communication networks and provides definitions of important assumptions; Section 3 introduces the proposed algorithm CAJA and the algorithm flow; Section 4 shows the simulation and the result analysis of the algorithm; and Section 5 provides a conclusion and a discussion of the outlook for the future.

## 2. System Model and Problem Modeling

### 2.1. System Model

This paper makes the following assumptions for a multi-user wireless communication system containing a central base station to facilitate the research.

1. Atypical multi-user wireless communication network has $M$ transmission channels accessible to every active user of the network, in addition to $N$ active communication users. Generally, a typical multi-user wireless communication network provides a sufficient number of channels available for access by users, so set $N < M$. To improve the success rate of the access channels in this network, in this paper, a central base station is added to the multi-user wireless network. This is used to unify the channel coordination and allocation for each user in the network. The structure schematic is shown in Figure 1.

2. All network users are equipped with broadband spectrum-perceived capability and have the ability to obtain the channel where current malicious interference is located through perception. The central base station serves as the decision-making center of the network and has the capacity to learn and make decisions. It can also coordinate users to access the transmission channel through a command signal, thereby enabling the combined anti-jamming of the system. All system users are within usable range of the central base station. None of the system users, nor the central base station, have prior information relating to external malicious interference.

3. Competition arises when all system users simultaneously use the same channel for transmission, and users in competition cannot successfully transmit data. For the central base station to be able to reasonably coordinate and uniformly allot internal user access channels to avoid competition during transmission, users of the system must have the ability to communicate with each other and exchange the perceived results. In addition, this paper sets the channel noise so it is not sufficient for affecting the communication performance of users within the system.

4. Communication time is divided into time slots of duration $T_s$ as the minimum time unit for continuous transmission, and interference time is divided in the same way. The time slots are further divided according to the responsibilities of each element in the system. The communication time slots of each system user are divided into observation sub-time slot $T_{obs}$ and action sub-time slot $T_{act}$, which are used for observing the external interference and the actions of other users in the system and communication transmission, as can be seen in Figure 2a. The communication time slots of the central base station are divided into decision sub-time slot $T_{dec}$ and learning sub-time slot $T_{tea}$, which are used for transmitting decision information, such as user information, channel selection information, and execution of algorithms, as can be seen in Figure 2b.

5. External malicious interference is set as high-power malicious interference in this paper, and the communication used by users is within the effective range of the malicious jammer. The external jammer senses the channel where each network user is located and selects the $J$ channel with the highest time slot utilization up to the current time slot—a single interference lasts for $L$ communication slots. The

interference style is Multi-channel Probabilistic Tracking jamming (MPT-jamming). Based on this, sweep interference is added for a specific user in the network to squeeze the central base station to coordinate and allocate the choice space of the user for channel access. The interference style is defined as compound intelligent jamming (CIJ).

6.  The constituent elements within this network, which include the central base station and each user, share communication time. Each time slot is strictly synchronized, and the perceived capability and perceived results for each user are kept consistent. The external malicious interference time is the same and is synchronized with the communication time within the network.

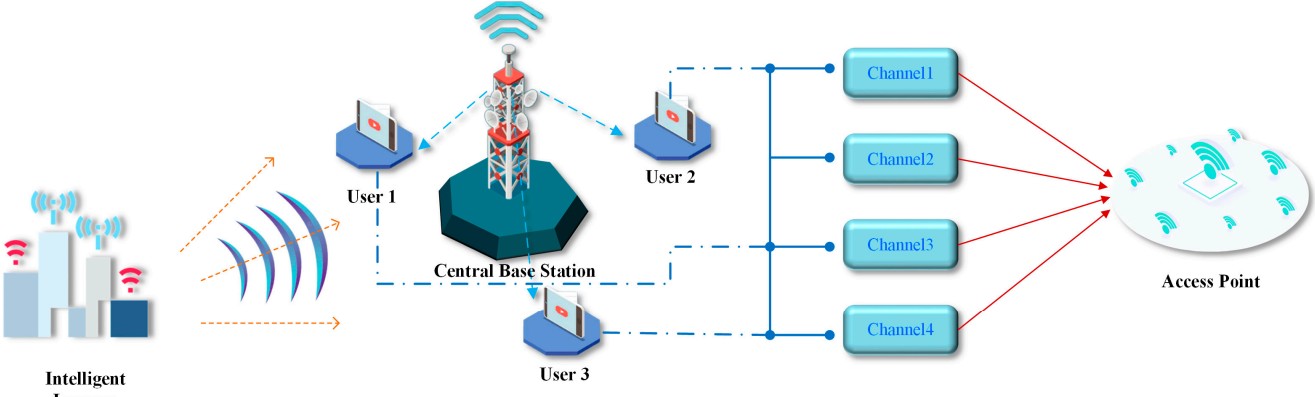

**Figure 1.** Schematic diagram of system structure (take $N = 3, M = 4$ as an example).

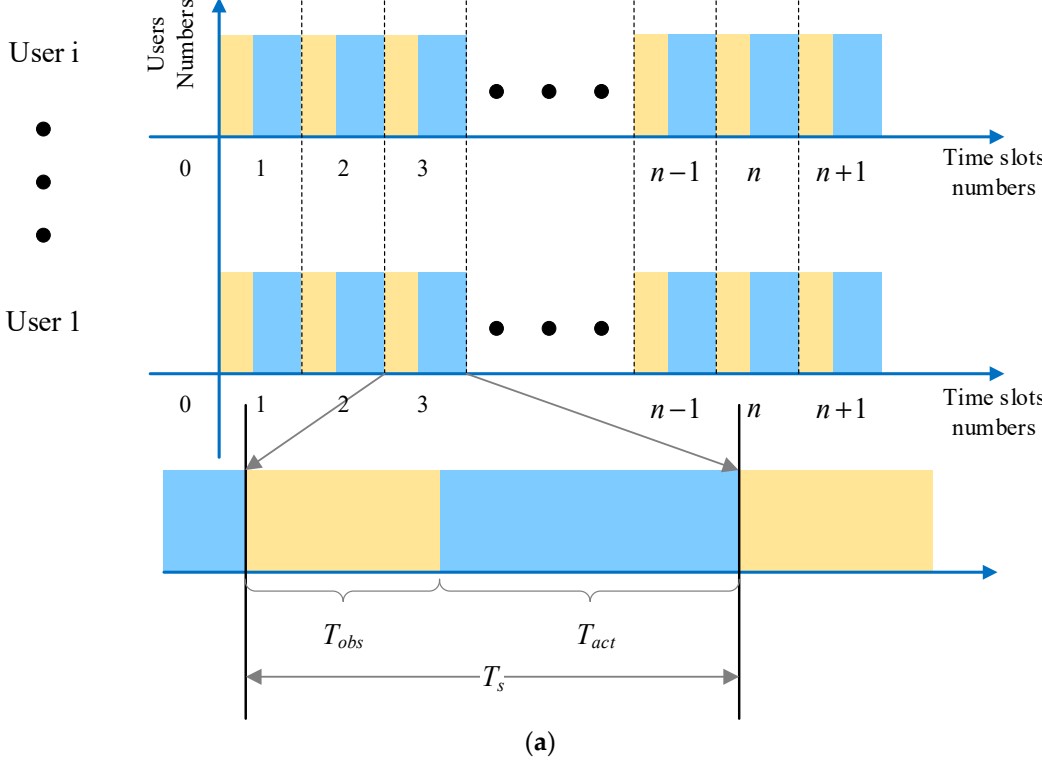

(a)

**Figure 2.** *Cont.*

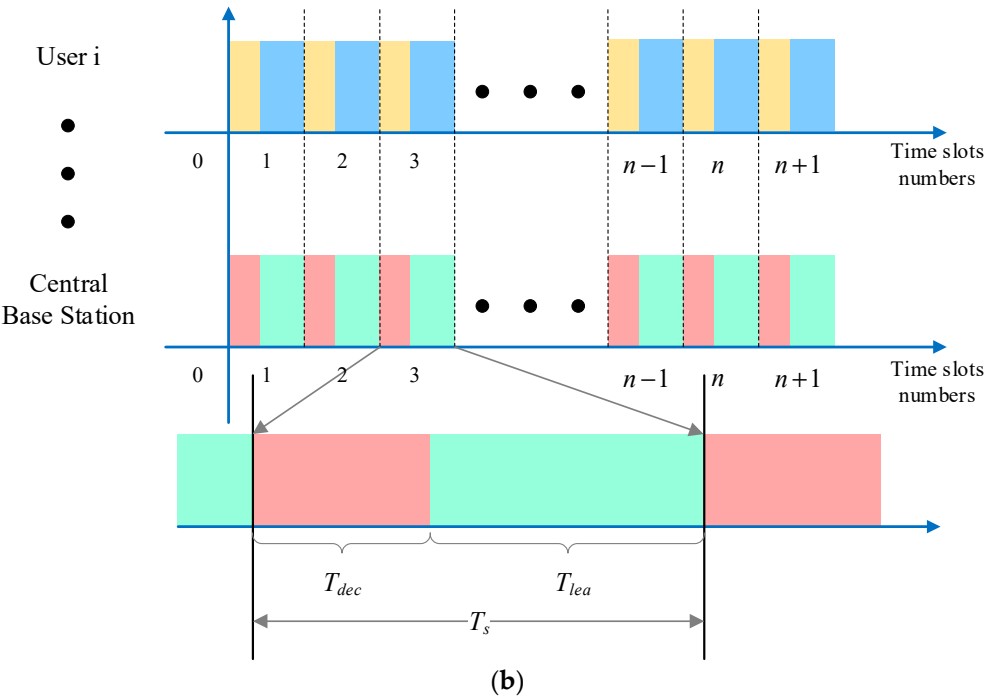

**(b)**

**Figure 2.** Schematic diagram of time slot structure: (**a**) schematic diagram of user time slot structure in the system; (**b**) schematic diagram of central base station time slot structure.

### 2.2. Problem Modeling

Regarding interference between users in the multi-user wireless communication network, the central base station is taken as the main body of the multi-user system, and single agent learning behavior is used for obtaining the optimal transmission strategy of the system. Therefore, the Markov decision process (MDP) is adopted for modeling the wireless communication network. The model can be expressed as a quadruple $\langle S, A, P, R \rangle$, where $S$ represents system state space, $A$ represents system action space, $P$ represents state transition probability, and $R$ represents reward function. In addition, based on the description of the number of users $N$ in the system, after the network is modeled as a single-agent system, the number of users is represented as the dimension of the system state space and the system action space. The following definition of a quadruple is used in this paper:

1.  State space $S$: this mainly reflects the current state of the channel that is provided by the environment. Taking a channel as an example, when it is idle, the channel status is *id* (idle), when it is jammed, the channel status is *ja* (jamming), and when it is occupied by users in the system, the channel status is *tr* (transmission). In the last two cases, the channel is defined as *bu* (busy) in this paper, which means the channel is occupied and busy. The communication decision of the system in the environment of external malicious interference is based on the perception result, or whether or not each channel is occupied. However, occupied channel information, which includes interference and information occupied by users in the system, is the knowledge that must be learned and applied by the system. Therefore, the system state space $S$ is defined as:

$$S \triangleq \{s | s = (b_1 \cdots b_z)\} \tag{1}$$

where $s = (b_1, b_2, \cdots, b_z)$ represents the set of crowded channels that is observed by the communication user, and $b_1 \cdots b_z$ subscript is the initial channel serial number in order from smallest to largest, $Z \in [1, M]$ and $Z \in R$. Therefore, $C_M^Z$ possible states exist in state space $S$.

2.   Action space $A$: in the multi-user scenario, the action space is a collection of actions that are independently chosen by each system user. According to the setting in this paper, the action of each communication user is choosing a single channel from the $M$ optional channels provided by the environment as a means of completing communication transmission. Therefore, the independent action subspace $A$ that belongs to each user in the joint action space $A_i$ of the system is the same, i.e., $A_1 = A_2 \cdots A_N$. Independent action space $A_i$ for a single user can be defined as follows.

$$A_i \triangleq \{ a_i | a_i \in \{1, 2 \cdots M\}\} \tag{2}$$

where $i \in \{1, 2 \cdots N\}$ represents the communication user serial number, and $a_i \in \{1, 2 \cdots M\}$ represents the channel serial number that is selected for transmission by communication user $i$, so the joint action $a$ of all communication users is a combination of the transmission channels of each communication user. The definition is as follows.

$$a = \{a_1, a_2 \cdots a_N\} \tag{3}$$

Therefore, joint action space $A$ is defined as:

$$A = A_1 \otimes A_2 \otimes \cdots \otimes A_N \tag{4}$$

where $\otimes$ denotes the Cartesian product. According to the above setting, a single communication user is only able to choose one channel for transmission at a single time. Therefore, there are $M$ possibilities in action subspace $A_i$, so there are $M^N$ joint actions in joint action space $A$.

3.   State transition probability $P : S \times A \times S' \to [0, 1]$ represents the probability of the set of agents transitioning to state $S$ after taking joint action $A$ in the channel state.

4.   Reward function $R$: the reward that is obtained by action $a_i$ taken by communication user $i$ in channel state $S$ and is dependent on the crowded transmission channel. A single transmission reward function $r$ for users in system $i$ is defined in the following way:

$$r(s, a_i) = \begin{cases} 1 & a_i \neq j_z \\ 0 & a_i = j_Z \end{cases} \tag{5}$$

where $i \in \{1, 2 \cdots N\}$ is the user serial number in the system and the instant reward for a single successful transmission by a single user is unit 1. When the user selects a channel for transmission, it is already occupied by other system users or interfered with externally, so there is no instant reward. All network users share instant reward function $r$, which means that the overall transmission instant reward function of system R is the sum of the instant reward functions of each system user.

$$R(S, A) = \sum_{i=1}^{N} r(s, a_i) \tag{6}$$

The system is transformed into a whole through the above modeling for solving the optimal decision of communication anti-jamming of the close-connected multi-user wireless communication network. Based on the Markov decision process (MDP), Q-learning is used for identifying the optimal strategy corresponding $\pi^*$ to the maximum cumulative return reward under the discount condition. The state-action function that corresponds to any strategy $\pi$ (also known as the Q value) is expressed as:

$$Q^\pi(S, A) = E\left\{\sum_{\tau=0}^{\infty} \gamma^\tau r_{r+\tau} | s_t = s, a_t = a, \pi \right\} \tag{7}$$

## 3. Central Anti-Jamming Algorithm Based on Improved Q-Learnings

A central base station is added to the network in this section for coordinating the channel access problems of system users as a means of effectively dealing with the problems of external malicious interference and inter-user interference in the multi-user wireless

communication network. A centralized anti-jamming algorithm based on improved Q-learning (CAJA) is proposed.

In order to reduce the competition phenomenon in multi-user wireless communication networks where internal users cannot transmit successfully, as they are crowding the same channel at the same time, the algorithm in this paper follows the sequential decision of the user number in the system; the central base station then learns the current state, one by one, based on the user number in the learning sub-time slot $T_{lea}$. For example, the central base station must only learn the current channel that is occupied by external interference when planning action $a_1$ of user 1 in the next time slot. At this time, state subspace $S_1$ of user 1 is as follows:

$$S_1 = \{s|s = (b_1 \cdots b_Z)\} \tag{8}$$

When the central base station is planning the action of user 2 in the next time slot, it must learn both the external and internal environments of the system, i.e., the result of the channel that was previously assigned by the central base station to user 1 for transmission. At this point, state subspace $S_2$ of user 2 is as follows:

$$S_2 = \begin{cases} S_1 & a_1 \text{ } The \text{ } selected \text{ } channel \text{ } is \text{ } not \text{ } interfered \\ S_1 \cup \{b_{a_1}\} & a_1 \text{ } The \text{ } selected \text{ } channel \text{ } is \text{ } free \end{cases} \tag{9}$$

where $b_{a_1}$ denotes the channel that is occupied by the user 1 selection transmission in the system when the elements in the state subspace of user 2 contain $\{b_1 \cdots b_Z, b_{a_1}\}$, in accordance with the above definition of state space, reordered according to the channel serial number, constituting a new state subspace $S_2$:

$$S_2 = \{s|s = (b_1 \cdots b_{a_1} \cdots b_Z)\} \tag{10}$$

Similarly, when the central base station begins learning actions $a_i$ of system users $i$ in the next time slot, the occupied channel perceived by user $i$ includes the intersection of the channel where the current interference is located and the result of the channel occupied by previous $i-1$ user, and the state subspace $S_i$ is as follows:

$$S_i = \{s|s = (b_1 \cdots b_{a_1} \cdots b_{a_{i-1}} \cdots b_Z)\} \tag{11}$$

The CAJA algorithm makes sequential decisions based on user serial numbers in the system in each iteration. Every user executes its own action $a_i$ in the joint action $a$ according to state subspace $S_i$ composed of the current perceived interference result and the channel information occupied by previous users and based on an action selection strategy that is executed by the central base station. It updates respective state $S_i'$ in the next time slot, while obtaining instant reward $r_i$ that belongs to that user at this time, before updating the value in the respective Q table based on Equation (12). This helps realize the optimization of the strategy, and the Q value update rule for each user is as follows:

$$Q_i(s,a) = \begin{cases} Q_i(s,a) + \alpha[r + \gamma \max Q_i(s',a') - Q_i(s,a)] & , s = s', a = a' \\ Q_i(s,a) & , else \end{cases} \tag{12}$$

where $i$ is the serial number of the system user, $\alpha$ is the learning factor, $r$ is the instant reward from the environment feedback when user $i$ performs an independent action $a$ under current state $s$, and y is the discount factor.

Once the Q value has been updated, the central base station updates the joint state of system $S_i'$ of the next time slot and actions $a_i'$ that are to be performed by the system following the completion of the allocation of every user access channel in the system and distributes them to each user as their own independent actions $a_i'$ through the command signal. Therefore, the tightly associated multi-user wireless system achieves a complete iteration under the overall allocation of the central base station, repeating this until the iteration is complete. Each complete time slot corresponds to a complete iteration of the

algorithm. The sequential coordination diagram of the central base station according to the Q table of each user can be seen in Figure 3.

The pseudocode of the centralized anti-jamming algorithm (CAJA), based on improved Q-learning, is as Algorithm 1:

---

**Algorithm 1:** CAJA

---

1: Initialisation: $\alpha$, $\gamma$; For any $S_i \in S$, $a_i \in A$, set in their respective current states $Q_i(S_i, a_i)$
2: **for** $t = 1, 2, \cdots, T$ do
3: Each user executes action $a_i$ according to joint action $a$ that is sent back by the central base station of the last time slot
4: Each user perceives the channel where current external interference occurs and obtains the current state of the external environment
5: The user perception results are shared to the central base station, which updates the status of each user according to the serial number
6: Each user receives instant reward $r_i$ from the environment by performing action $a_i$ in their respective state $S_i$
7: The central base station makes sequential decisions based on user numbers in the system, updating their respective Q values according to Formula (12)
8: The central base station makes a unified summary and calculates the overall Q value of the system at this time using the following formula:

$$Q(S, a) = \sum_{i=1}^{N} \left\{ Q_i(s, a) + \alpha \left[ r_i + \gamma \max Q_i(s', a') - Q_i(s, a) \right] \right\} \qquad (13)$$

9: The central base station assigns the next time slot to each user based on the following action selection strategy $a'_i$:

$$a'_i \begin{cases} \pi^\varepsilon(s_{t+1}) = \mathrm{argmax}_{a_i \in A} Q_i(s_i, a_i) & \text{When the probability is } 1 - \varepsilon \\ \forall a \in A & \text{When the probability is } \varepsilon \end{cases} \qquad (14)$$

10: Update the respective states and actions: $S_i = S'_i$, $a_i = a'_i$; the central base station distributes the command signal to the respective users in the system;
11: $t = t + 1$
12: **end for**

---

The specific procedure is as follows:

From the current perception results, each system user is then assigned the channel that is accessed by each user by the central base station uniformly and initializes their respective exclusive Q-table by user number, based on the results shared by each user. The following actions are then completed in each complete time slot: system users perform their own independent actions $a_i$ according to joint actions $a$ that were sent back by the central base station in the previous slot (line 3); in observation sub-slot $T_{obs}$, each user perceives the current external state (line 4) and synchronously shares the perception results with the central base station; in learning sub-time slot $T_{lea}$, the central base station updates the state of each user (row 5) according to the user number and the execution of independent actions $a_i$ by each user, calculating the instant reward $r_i$ (line 6) for each user to perform their respective actions in the current state, updating the Q value in the exclusive Q table of each user (line 7) and aggregating the Q value of the system as a whole (line 8); in decision sub-time slot $T_{dec}$, the central base station coordinates and assigns independent actions $a'_i$ for each user in the next time slot, based on Equations (13) and (14) (line 9); the central base station ultimately distributes the decision result of joint state $S'$ and joint action $a'$ to each user as independent state $S'_i$ and independent action $a'_i$ through command signals (line 10), pending the execution of the action assigned in this time slot at the start of the next iteration in execution sub-time slot $T_{act}$ by each user. A complete iteration is completed at this point (line 11). The flow of decisions within the time slot can be seen in Figure 4.

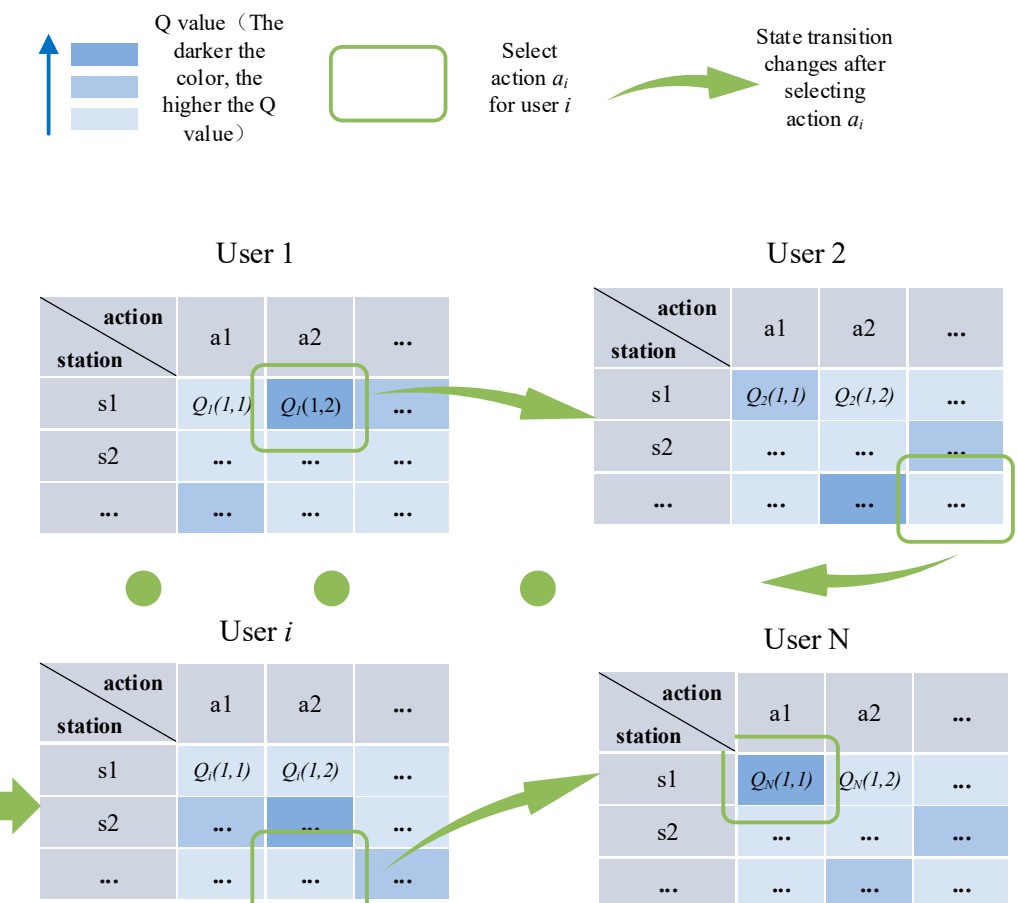

**Figure 3.** Sequential Q diagram.

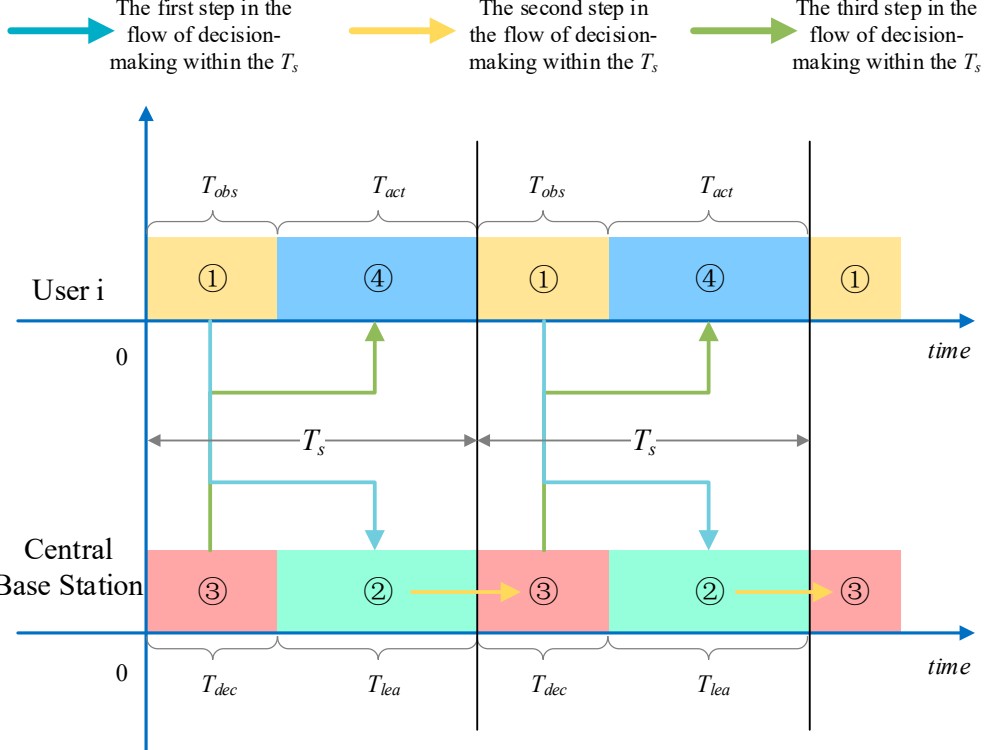

**Figure 4.** Schematic diagram of decision flow in a time slot.

## 4. Simulation Results and Experimental Analysis

In this section, the proposed centralized anti-jamming algorithm (CAJA) that is applicable to the tightly correlated multi-user wireless communication network is simulated on the MATLAB platform. This will verify the performance index of the proposed algorithm in anti-jamming communication under the dual influence factors of external and internal interference.

### 4.1. Parameter Settings

Relevant parameters in the simulation are set as Table 1:

**Table 1.** Simulation parameter settings.

| Parameter | Numerical Value |
| --- | --- |
| Number of users $N$ | 3 |
| Number of available channels $M$ | 10 |
| Length of communication time slot $T_s$ | 0.3 ms |
| Observation sub-time slot $T_{obs}$ | 0.1 ms |
| Decision sub-time slot $T_{dec}$ | 0.1 ms |
| Action sub-time slot $T_{act}$ | 0.2 ms |
| Learning sub-time slot $T_{lea}$ | 0.2 ms |
| Selective number of tracking jamming $J$ | 2 |
| Number of continuous time slots of follower jamming $L$ | 3 |
| Discount factor $\gamma$ | 0.6 |
| Learning rate factor $\alpha$ | 0.8 |
| Greedy factor $\varepsilon$ | $1/\sqrt{t}$ |
| Total number of time slots $N_s$ | 2000 |

For the effective evaluation of the performance of the centralized anti-jamming algorithm (CAJA) that is proposed in this paper, the following two anti-jamming schemes are set up for comparison:

Independent Q-learning (IQ): each user in the network perceives and performs Q-learning independently based on the perception results. No interaction occurs between network users, and information is not shared, with each user independently executing the decision.

Orthogonal frequency hopping (OFH): each user selects the transmission channel according to the mutually orthogonal frequency-hopping pattern that is agreed upon in advance. According to the hopping pattern, no mutual interference exists between network users, and there is no "competition" phenomenon, where two or more users share the same channel.

### 4.2. Analysis of Simulation Results

Both the proposed algorithm and the comparison algorithm are simulated using the MATLAB platform.

The first 30 time slots (0–30 time slots) of the close-related multi-user wireless communication network as it begins working were obtained based on the parameter settings from the previous section, under the distribution and coordination of the central base station. The time slot-channel selection diagram of three users can be seen in Figure 5.



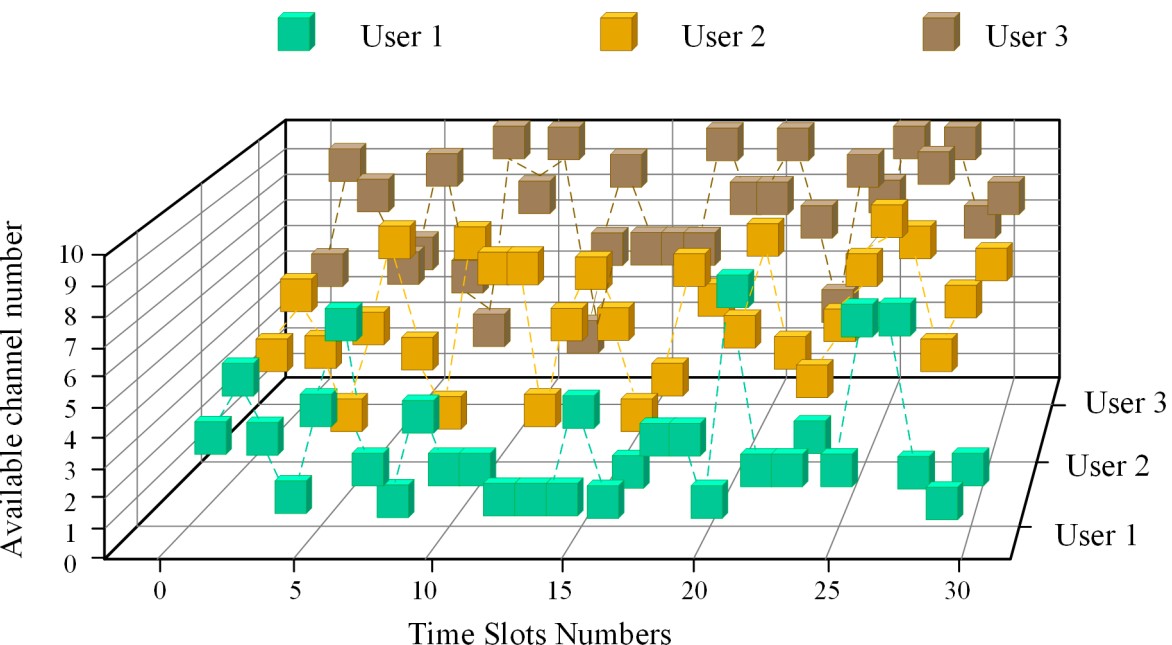

**Figure 5.** Schematic diagram of 3-user system time slot-channel selection cube, including the central base station.

The intelligent tracking interference is based on the channel selection of users in past time slots, where the probability of selecting an interfering channel per time slot changes dynamically according to the channel selected by the communicating party's communication. The implementation of composite intelligent interference (CIJ) will sweep the interference to user 3, i.e., user 1 and user 2 within the network face an intelligent tracking-type interference threat. Figure 6 shows the thermal diagram of the implementation probability time slot-channel distribution of tracking sub-interference in composite intelligent interference.

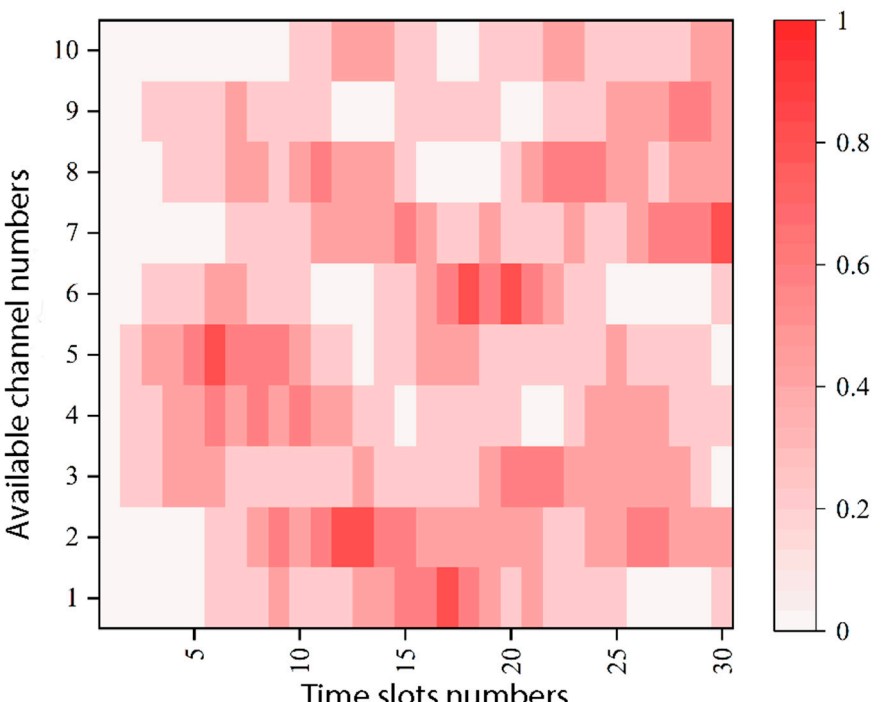

**Figure 6.** Thermal diagram of intelligent tracking sub-jamming time slot-channel probability distribution.

The system-jamming reality diagram was drawn by combining Figures 5 and 6 with the characteristics of intelligent tracking sub-interference and swept frequency sub-jamming for user 3 in compound intelligent interference. This can be seen in Figure 7.

With the double threat of intelligent tracking-type jamming and sweep jamming in the compound intelligent jamming environment, the multi-user wireless communication network that contains a central base station has a poorer transmission effect than that of the ordinary multi-user wireless communication network at the beginning of the work. In this case, the user channel selection cubes in the system collide with the jamming cubes more often.

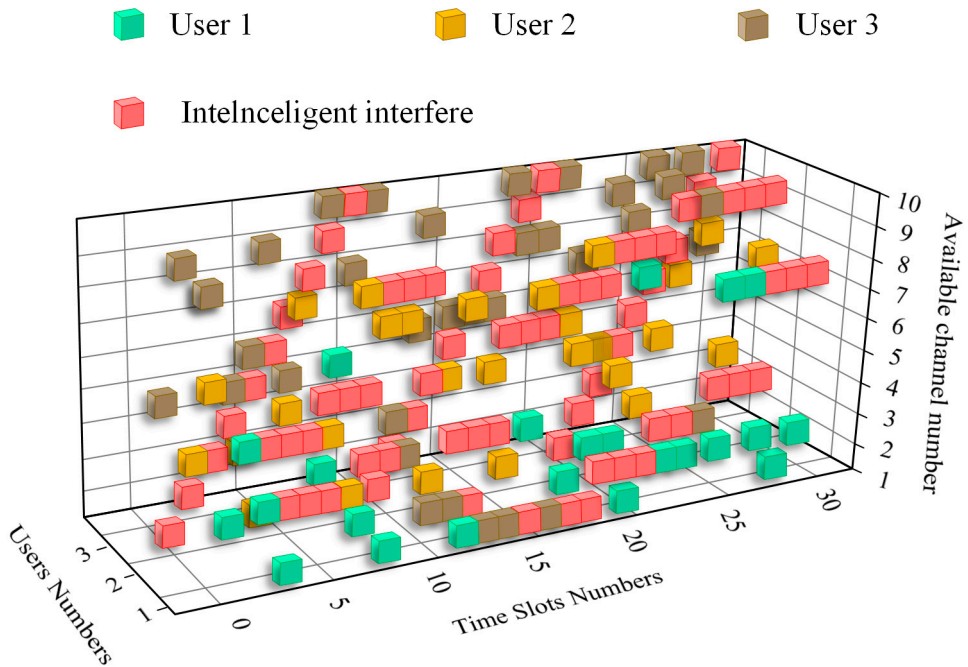

**Figure 7.** Real interference-communication diagram of a 3-user system with a central base station.

However, due to the channel selection coordination of the central base station for users in the system, the central anti-jamming approach avoids "competition" between network users that will result in transmission failure for relevant users in that time slot by virtue of the advantage of sequential decision making. This is conducive to reducing transmission failure caused by internal factors in the early stage of the system. In other words, the reason for transmission of user failure within the system in a particular time slot is external compound interference, and this situation is favorable for enabling the wireless communication network to take advantage of the technical advantage of reinforcement learning, which allows it to search for optimal solutions through errors. The central base station learns the feedback from the environment, acquiring knowledge and using it as a basis for better coordinating the channel that is accessed by each user in the wireless communication network in the next time slot. This gives full play to the learning advantage of reinforcement learning during unknown interference.

Therefore, the jamming-communication reality of the wireless communication network in the middle of the iteration (1000th to 1030th time slots) is chosen as a comparison. The schematic of this situation can be seen in Figure 8.

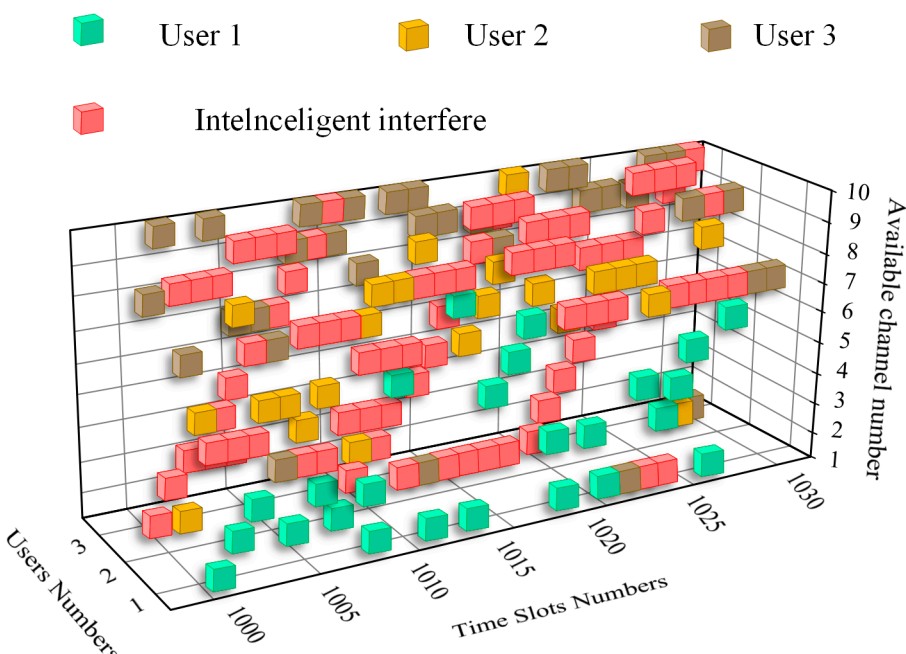

**Figure 8.** Schematic diagram of system channel selection-interference cubes in the middle of the iteration.

A longitudinal comparison of Figures 7 and 8 shows that the multi-user wireless communication network with a central base station is better protected from sweep interference and intelligent tracking interference threats in the composite intelligent interference at the middle of the set iteration time than at the start of the communication. In comparison to the loosely coupled multi-user wireless communication network, with relatively fixed anti-jamming strategies for each system user during the convergence state, the multi-user wireless communication network with a central base station does not possess a fixed anti-jamming strategy. The channel selection strategy for each user is more flexible under central base station unified coordination, regardless of whether user 1 and user 2 face an intelligent tracking-type interference threat, or user 3 faces conventional sweep jamming.

When internal users are faced with a complex intelligent interference threat, the defined intelligent tracking interference becomes more random as a result of the flexible channel selection strategy. At this point, randomly implemented tracking interference is prone to conflict with the hopping channel selection strategy of the user. This then causes transmission failure for a certain user in a certain time slot, making the randomness event unavoidable. In addition, a closer observation of Figure 8 reveals that channel selection and interference collision during transmission is not as small as that envisioned when user 3 faces only a conventional sweep jamming threat. This is because conventional sweep interference is more regular and can more easily be perceived by users and mastered by the central base station in comparison to intelligent interference. Therefore, the transmission income of user 3 should have been the best in this system. However, due to the characteristics of the sequential decision making of the central base station and the number-based coordination of access channels, the first coordinated user has a larger alternative space for action selection, and the second has a smaller space for action selection. In addition, the external malicious interference implements conventional frequency sweep interference for user 3, further reducing the action selection space of the central base station for it. User 3 should have the best interference-avoiding effect, but possesses has many transmission failure slots as a result of non-self-factors caused by algorithm inadequacies.

To make a more intuitive horizontal comparison between the proposed algorithm and the system transmission performance under the same conditions, this paper defines the average user transmission income of the system as follows:

$$r_{ave}(t) = \sum_{i=1}^{N} E_i(t) / N \cdot W \tag{15}$$

where $r_{ave}(t)$ is used for calculating the average transmission income of users up to time slot $t$ in a closely interconnected multi-user wireless communication system. $E_i(t)$ represents transmission income from the environment for user $i$ up to time slot $t$; $\sum$ is the cumulative transmission income of all system users up to time slot $t$; $N$ is the number of network users, which this paper sets as 3; $W$ is the number of independent algorithm runs, which is executed once in each unit time slot. Based on the definition of system model transmission income, the income from the environment of a single user in the system at a certain time slot is unit 1, so $E_i(t)/W$ is the ratio of the successful transmission time slot of user $i$ to the current accumulated time slot as of time slot $t$. A comparison is performed between the proposed algorithm, the comparative anti-jamming algorithm, and the scheme transmission income performance, which is shown in Figure 9.

Following the observation of Figure 9a, it can be seen that the proposed centralized anti-jamming CAJA has the best average user transmission income, followed by the independent Q-learning anti-jamming algorithm. The orthogonal frequency hopping anti-jamming scheme exhibits the worst performance. At the 500th time slot, the average transmission income value is 0.749 for CAJA users, 0.678 for IQ users, and 0.592 for orthogonal frequency hopping users. The proposed algorithm improves the average income by approximately 10.49% in comparison to IQ users and approximately 26.47% in comparison to the traditional orthogonal frequency hopping scheme.

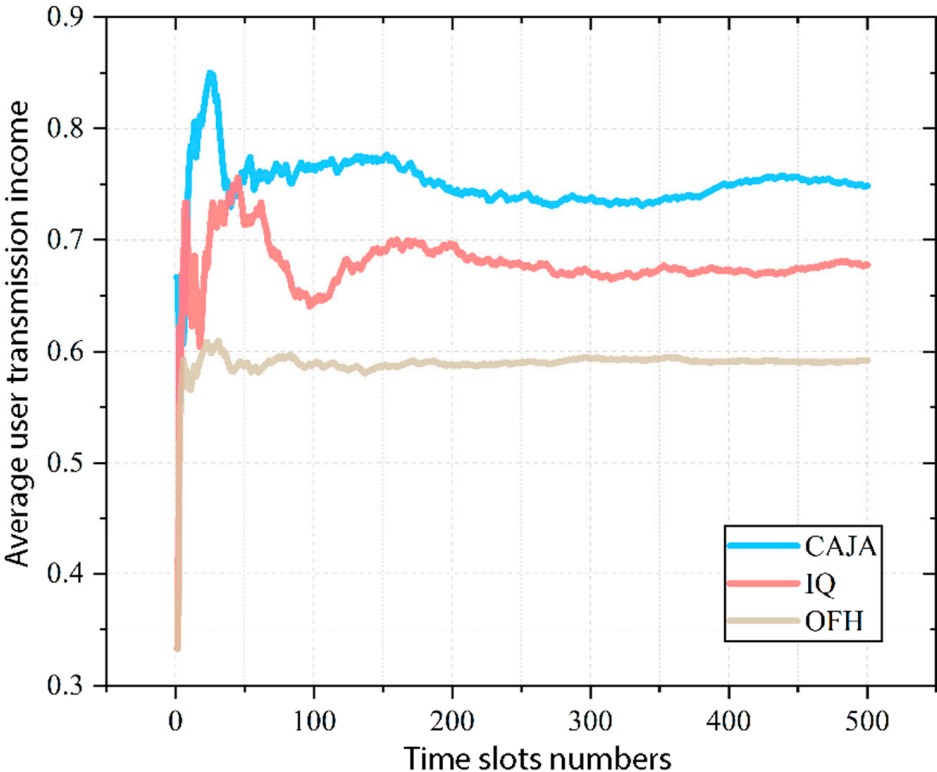

(**a**)

**Figure 9.** *Cont.*

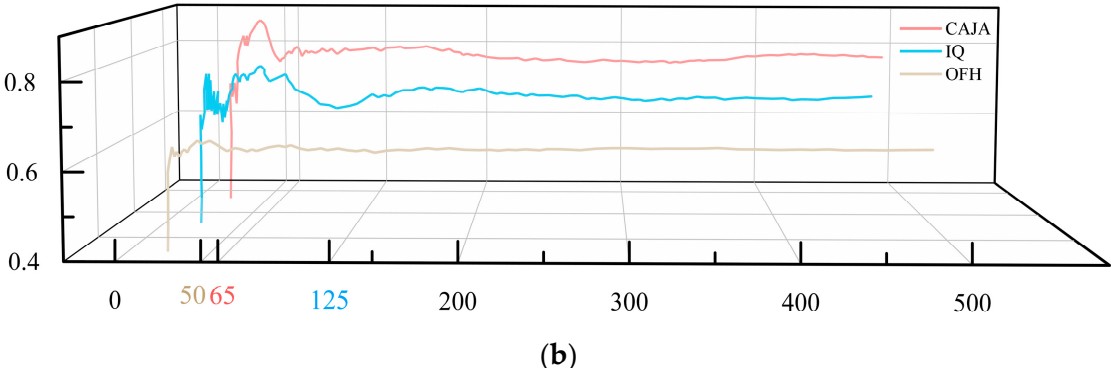

**(b)**

**Figure 9.** Comparison diagram of average transmission income.

After performing an analysis of the characteristics of the three aforementioned anti-jamming schemes in Figure 9a, the transmission performance of the central anti-jamming algorithm was found to be the best. However, due to the complex intelligent interference characteristics, user 3 targeted by the sweep interference exhibits compressed action selection space due to the sequential decision of the central base station, and it collides with the sweep interference, which results in a transmission failure in a certain time slot. This then causes overall system transmission income fluctuations. Due to the independent perception and decision of each system user, it is not possible to completely avoid interference between users. The average transmission income of the system users is limited by the influence of interference, which results in an inferior transmission performance compared to the wireless communication network that includes the central base station. However, the traditional orthogonal frequency-hopping method is able to completely escape interference between users, but malicious external interference cannot be completely avoided. This is particularly evident in the complex intelligent jamming environment of this paper where anti-interference ability is limited, and transmission performance is mediocre.

By analyzing the above three anti-interference schemes in Figure 9b, we find that the OFH method has the fastest convergence speed, but the lowest income, and the CAJA method has the highest average income, although its convergence speed is slower compared with that of the OFH method. Compared with OFH and CAJA, the convergence speed of the IQ method is slower than that of OFH, and the income is lower than that of CAJA, so the results are relatively mediocre.

## 5. Conclusions

To further improve multi-user wireless communication network adaptability in a complex interference environment, a central base station for unified coordination and allocation of access channels for each user in the network was added, based on loosely coupled multi-user wireless communication network architecture. This also helped effectively avoid complex external interference and mutual interference between network users, improving the overall transmission income of the system in an interference environment. In addition, the central base station was taken as the decision-making body of the system. The system model and external environment were modeled according to the single-user Markov decision process of the multi-user wireless communication network, and the anti-jamming algorithm of the system was designed using improved Q-learning. The proposed algorithm used the perceived result shared by all system users as one of the decision bases of the central base station and made decisions to be executed sequentially based on the user number in the system. At the same time, the action of the previous user was taken as the decision environment of the next user, and transmission was completed in the form of a Q-table. After the central base station completed the action decision of all users in the next time slot, it distributed it to each user in the form of their respective independent actions through the command signal. Simulation results found that the proposed CAJA algorithm improved the transmission income by approximately 10.49% over

the independent Q-learning multi-user anti-jamming algorithm and approximately 26.47% over the traditional orthogonal frequency hopping anti-jamming scheme in the composite intelligent interference environment set up in this paper. The next step will be attempting to combine deep learning and reinforcement learning to help the communication party fully learn the external environment and interference characteristics. Reinforcement learning will be added to deep learning as a means of helping the system realize the optimal communication decision under the interference environment while further enhancing the overall communication network performance.

**Author Contributions:** Conceptualization, Y.N., C.C. and B.W.; methodology, Y.N.; validation, B.W.; formal analysis, Y.N., C.C. and B.W.; investigation, Y.N., C.C. and B.W.; resources, Y.N., C.C. and B.W.; data curation, Y.N., C.C. and B.W.; writing—original draft preparation, B.W.; writing—review and editing, Y.N. and C.C.; supervision, Y.N. and C.C.; project administration, Y.N.; funding acquisition, Y.N. All authors have read and agreed to the published version of the manuscript.

**Funding:** This research was funded by the National Science Foundation of China (NSFC grant: U19bB2014).

**Data Availability Statement:** The data presented in this study are available upon request from the corresponding author. The data are not publicly available due to privacy constraints.

**Conflicts of Interest:** The authors declare no conflict of interest.

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
