# Peer review of "A Centralized Multi-User Anti-Composite Intelligent Interference Algorithm Based on Improved Q-Learning"

_electronics, doi:10.3390/electronics12081803_

Round 1
Reviewer 1 Report
The paper entitled "A Centralised Multi-User Anti-Composite Intelligent Interference Algorithm Based on Improved Q-Learning" presents a centralised anti-jamming algorithm based on improved Q-learning for multi-user wireless communication networks. The proposed algorithm models the system using a single-user Markov decision process and considers the impact of external malicious interference and inter-user interference on the communication system to improve overall system transmission income. The manuscript is well-written and structured, with a clear introduction and explanation of the proposed algorithm. The simulation results presented in the manuscript are well-documented, and the authors have provided a detailed analysis of the algorithm's performance. The results show that the proposed algorithm significantly improves overall system transmission performance compared to existing multi-user independent Q-learning anti-jamming algorithms and traditional orthogonal frequency-hopping schemes.
However, there are some minor issues that need to be addressed.
- The authors need to define the abbreviations used in the manuscript before using them.
- The authors have provided a comprehensive literature review but also needs to add some most recent research, e.g., “Ultrafast and wideband microwave photonic frequency-hopping systems: A review. Applied Sciences, 2020.”
- Some of the sentences are too long and complex, making it difficult to understand the intended meaning.
- The paper lack of real-world experimental validation of the proposed algorithm and the assumption of perfect perception of malicious interference by all network users.
- Please make the figure fonts and style consistent in the paper. For instance, in fig5, it lacks of x label; In fig7/8, “time slots numbers” needs to be capitalized.
Based on my assessment, I recommend accepting the manuscript for publication in Electronics journal after addressing the minor issues raised.
Reviewer 2 Report
Dear Authors,
This paper proposed a central anti-jamming algorithm (CAJA) based on improved Q-learning to reduce the impact of inter-user interference, external malicious interference and improve the transmission income.
The major contribution of the paper is the application of Q-learning to improve overall system transmission income and avoid external malicious interference.
Following are my comments:
Please explain the drawbacks of references [13] and [14]. How is your work different from these references?
The concept of perception is vague. Please explain what the network users perceive to obtain the channel where malicious interference is located.
Please explain how the proposed algorithm can mitigate the impact of channel noise.
In independent Q-learning, can the network users share the same channel? Please explain.
Please explain the results in figures 5 and 6 in detail. The current discussion is vague.
What does "Intelncligent interfere" mean in figures 7 and 8. Do you mean intelligent interference?
What version of Matlab is used for the simulations? Did you use any toolbox in Matlab? Please explain.
Please make sure the spacing is consistent between the sentences.
Equations and several abbreviations within the sentences are not appropriately aligned.
Replace the word "chapter" with "section".
Please check the spacing between the words carefully.
Reviewer 3 Report
The paper "A Centralised Multi-User Anti-Composite Intelligent Interference Algorithm Based on Improved Q-Learning" proposes a Central Anti-Jamming Algorithm (CAJA) based on Improved Q-Learning to solve the communication problems faced by multi-user wireless communication networks, considering external complex malicious interference, and reducing the dual factors limiting the quality of wireless communication, as well as the effects of inter-user interference within the network and external malicious interference on the communication system, so as to improve the transmission revenue of multi-user wireless communication.
The simulation results presented in this paper showed that the proposed CAJA algorithm improved the transmission revenue by about 10.49% over the independent multi-user interference protection algorithm with Q-learning and by about 26.47% over the conventional interference protection scheme with orthogonal frequency hopping in a composite intelligent interference environment, which has been outlined in this paper.
As a quick note, the paper states that the next step will be to attempt to combine Deep Learning and Reinforcement Learning to fully learn the characteristics of the environment and interference, which in turn will help implement the optimal communication solution in interference environments, while improving overall communications network performance. The question of what prevented the combination of Deep Learning and Reinforcement Learning methods from being used in the first place needs to be clarified.
For specific improvements should be considered to combine Deep Learning and Reinforcement Learning to fully explore the characteristics of the environment and interference, which in turn will help implement an optimal communications solution in the interference environment, while improving the overall performance of the communications network.
The work is well done.
Round 2
Reviewer 2 Report
Dear Authors,
Thank you for answering all my queries.
The symbols within the text are not properly aligned. I hope the editorial team can take care of this issue.